# Mortality in Pedigrees with Acute Intermittent Porphyria

**DOI:** 10.3390/life12122059

**Published:** 2022-12-08

**Authors:** Rochus Neeleman, Kyra Musters, Margreet Wagenmakers, Sophie Mijnhout, Edith Friesema, Eric Sijbrands, Janneke Langendonk

**Affiliations:** 1Erasmus MC, Department of Internal Medicine, Center for Lysosomal and Metabolic Diseases, Porphyria Expertcenter Rotterdam, University Medical Center Rotterdam, 3000CA Rotterdam, The Netherlands; 2Isala Clinics, Department of Internal Medicine, 8025AB Zwolle, The Netherlands

**Keywords:** acute intermittent porphyria, acute hepatic porphyria, hydroxymethylbilane synthase, standardized mortality ratio

## Abstract

High mortality rates have been reported in historical cohorts of acute intermittent porphyria (AIP) patients. The mortality associated with (hydroxymethylbilane synthase) *HMBS* variant heterozygosity is unknown. This study estimates all-cause mortality in pedigrees with *HMBS* gene variants that cause AIP. We collected data on the lifespan of individuals in Dutch AIP pedigrees and performed analyses using the family tree mortality ratio method. This gave us standardized mortality ratios for these pedigrees compared to the Dutch general population as a primary outcome. Between 1810 and 2017, the overall mortality in these pedigrees was identical to that of the general Dutch population: (SMR 1.01, *p* = 0.441). However, compared with the general population the SMR was significantly higher in women aged 45–64 years (SMR 1.99, *p* = 0.00003), which was based on excess mortality between 1915 and 1964 (SMR 1.94, *p* < 0.00002). In men aged 70–74 years, the SMR was 1.55 (*p* = 0.0021), based on excess mortality that occurred between 1925 and 1964 (SMR 1.92, *p* = 0000000003). Overall, mortality from *HMBS* variant heterozygosity was not increased compared with the general population. Severe excess mortality occurred in young women and old men between 1915 and 1964. Heterozygotes reached a normal lifespan during the past half-century, in parallel with disease awareness and the prevention of new attacks through family counselling.

## 1. Introduction

Acute intermittent porphyria (AIP) is an inherited metabolic disorder of the heme biosynthesis pathway. Heterozygous variants in the hydroxymethylbilane synthase (*HMBS*; MIM# 609806) gene diminish the enzyme activity of porphobilinogen deaminase (PBGd) [1]. Consequently, this leads to the accumulation of the potentially neurotoxic heme precursor delta-aminolevulinic acid (ALA), which is highly increased in plasma and urine during an acute porphyric attack [2]. The clinical features of an acute attack include severe abdominal pain, psychiatric symptoms, hyponatremia, and a neuropathy that can progress to respiratory failure and death [3]. Only a very small subset of patients suffer from recurrent (≥4 per year) acute porphyric attacks and require frequent hospitalizations. This results in a high disease burden, high mortality, and a low quality of life [4,5].

The prevalence of variants in the *HMBS* gene is 1 in 1675 in a healthy population [6], which is high compared to the prevalence of attacks. Moreover, the penetrance of acute attacks in the general population with likely pathogenic variants is estimated to be around 1% of heterozygotes [7]. Thus, most latent AIP heterozygous gene carriers remain asymptomatic during their lifespan (>90%). However, in families with index patients with known pathogenic variants, the penetrance of acute attacks has been observed to be between 10% and 22%, suggesting that environmental and genetic factors unrelated to *HMBS* play an important role in AIP’s penetrance [8]. Known attack-provoking factors include porphyrinogenic drugs, fasting, infections, smoking, and hormonal fluctuations during the menstrual cycle. Moreover, attacks are more prevalent in women than in men. In addition to acute attacks, latent AIP heterozygous gene carriers are at increased risk of long-term complications such as hypertension, chronic kidney disease, and hepatocellular carcinoma [9].

AIP is part of the broad differential diagnosis of recurrent abdominal pain and is often misdiagnosed. Acute porphyria attacks can be life-threatening when the attack is not recognized and treatment with heme arginate is not started. Historical cohorts from before the 1970s, mostly including patients requiring hospitalization, reported high mortality rates following porphyric attacks—ranging between 21 and 59% [10,11,12,13,14].

However, the mortality rate of carrying a known pathological variant in the *HMBS* gene without selection for acute attacks—including carriers with latent or inactive disease—is unreported. This information is valuable, as most of the heterozygous *HMBS* gene variant carriers never experience an acute porphyric attack. In the present study, the mortality rate of *HMBS* variants was assessed in large pedigrees with known pathogenic *HMBS* variants.

Mortality was assessed without bias with respect to the occurrence of attacks, but the individuals were at risk since they were from families with at least one active (i.e., index) patient who presented with an attack. Mortality was established irrespective of morbidity, as morbidity is impossible to reliably determine in pedigrees with several generations.

## 2. Materials and Methods

### 2.1. Data Collection

Available pedigrees and patient data collected from the Porphyria Expertcenter Rotterdam (Erasmus MC) were used (1960–2017) as the starting point for subject selection. All patients had been either symptomatic or detected by family counselling. From the pedigrees and patient data, we included data from all individuals with a definite diagnosis of AIP. This was based on either the presence of a known pathological *HMBS* variant or a decrease in enzyme activity defined as PBGd activity (normal PBGd activity in erythrocytes: >64 pmol/mg protein/h; normal PBGd enzyme activity in lymphocytes: >72 pmol/mg protein/h (since September 2006) or > 80 pmol/mg protein/h (before September 2006)). Moreover, if two individuals shared the same variant, all ancestors in the connecting lineages (i.e., paternal or maternal bloodlines) could be included as heterozygotes, without medical information or diagnostic test results. Appendix A provides more insight into this procedure. Obligate heterozygotes were identified based on the examination of family lineages. These individuals all had an a priori Mendelian probability of 1.0 of being heterozygous for the variant. All first-degree relatives (i.e., untested siblings and children) of these individuals were included. The data of complete sibships along the lines of transmission had a probability of 0.5 of being heterozygous. The primary analyses were performed on these sibships to avoid the deselection of severe mortality risk in the past that could not be expressed in the present generations [15]. All individuals who died before the age of one year were excluded from the analysis; this eliminated the bias from expected underreporting of neonatal deaths in the official records, most likely during the early 19th century.

We performed genealogical searches in publicly available databases of official Dutch archives for all included individuals to confirm their dates of birth and death. The Dutch governmental organization CIBG (http://www.cibg.nl, accessed on 6 February 2019) offers transparent and reliable Dutch population data, having collected data on the birth and death dates of all Dutch inhabitants since the 19th century.

### 2.2. Family Tree Mortality Ratio (FTMR) Method

The ratio of observed-to-expected mortality was calculated as a standardized mortality ratio (SMR). The mortality rates in the family trees (i.e., observed mortality) were compared to the mortality rates of the Dutch population (i.e., expected mortality) and standardized for age, sex, and calendar period, as previously described [15]. The expected mortality was calculated by multiplying the total number of person-years lived by the study population in each calendar period per sex and age category by the age- and sex-specific mortality rates of the Dutch population for each calendar period between 1810 and 2017. These population rates were obtained from the Dutch governmental office Statistics Netherlands (https://www.cbs.nl; accessed on 6 February 2019) using the computer program Person-Years [16]. The 95%CI of the SMR was calculated assuming a Poisson distribution of the observed number of deaths and by using exact limits. We analyzed the mortality for 10-year intervals between 1810 and 2017 and for the following age ranges: 0 < 1 year, 1 to 4 years, 5 to 9 years, 10 to 14 years, 15 to 19 years, 20 to 24 years, 25 to 29 years, and so on, until the last category of 85 to 109 years.

## 3. Results

### 3.1. Genealogical Searches

The dataset initially contained 1935 persons from 46 AIP pedigrees. A total of 1308 persons were excluded based on the absence of a definitive or inferred AIP diagnosis. Another 100 persons were excluded for missing essential information on their year of birth or death. Additionally, three persons who were born before 1810 were excluded, as governmental population data in the Netherlands are available only after 1810. Finally, one person was excluded as stillborn. This resulted in a cohort with 523 persons (46 pedigrees, ranging from 1 to 45 persons, and 29.680 person-years), as shown in Table 1. An overview of all the different *HMBS* variants identified in this cohort and their prevalence is presented in Table 2.

### 3.2. Standardized Mortality in Different Time Periods

Between 1810 and 2017, 355 deaths were observed in 29,680 person-years of 523 individuals; this corresponded to an SMR of 1.01 (95%CI [0.91 to 1.12], *p* = 0.441). Adjusted for differences in the distributions of age categories and calendar periods, women did not have a significantly different mortality risk than men (Poisson regression rate ratio 0.85, 95%CI [0.69 to 1.05], *p* = 0.138). Figure 1a–c presents the SMR in this Dutch AIP pedigree cohort compared to the general population in different decades between 1810 and 2017, separated by sex. Compared to the general population significant excess mortality was observed in women between 1915 and 1964 (SMR 1.94, 95%CI [1.48 to 2.50], *p* < 0.000002), with a peak between 1945 and 1954 (SMR 2.72, 95%CI [1.56 to 4.44], *p* = 0.0004; Figure 1a). In men, the SMR was significantly increased between 1925 and 1964 (SMR 1.92, 95%CI [1.58 to 2.32], *p* = 0.0000000003), with a peak between 1935 and 1944 (SMR 2.58, 95%CI [1.61 to 3.9], *p* = 0.0001; see Figure 1b).

### 3.3. Mortality Risk According to Age Category

Figure 2a–c shows the SMRs per age category between 1810 and 2017 compared to the general population. In women between 45 and 64 years of age, the SMR was significantly increased (SMR 1.99, 95%CI [1.44 to 2.26], *p* = 0.00003; Figure 2a). In men aged 70–74 years, the SMR was 1.55 (95%CI [1.02 to 2.25], *p* = 0.021; Figure 2b).

## 4. Discussions

In a Dutch AIP pedigree cohort comprising persons living between 1810 and 2017, the overall mortality compared to the general population was not increased. However, severe excess mortality took place in women between 1915 and 1964 and in men between 1925 and 1964. Mortality from *HMBS* heterozygosity was greater in women than in men; women aged between 45 and 64 years had significant excess mortality, whereas only older men aged between 70 and 74 years had excess mortality. This has important implications for clinical genetics counsellors. More than 90% of *HMBS* heterozygotes remain asymptomatic lifelong, as the penetrance of *HMBS* gene variants is low [6,7], and that subgroup seems to have normal life expectancy based on these results.

The peak in mortality between 1915 and 1964 might, in hindsight, be explained by several currently known provoking triggers. First, the advent of the modern pharmaceutical industry and the subsequently widespread availability of pharmaceuticals just between the World Wars (e.g., sulfonamide and phenobarbital) could explain excess mortality occurring after 1925 [17]. The prescription of such drugs may have resulted in lethal porphyric attacks at a time when effective therapy for AIP was unavailable. Second, during the worldwide economic depression of the 1930s, there was an extended period of limited food supplies in large parts of the Netherlands. Malnutrition is likely to have increased the odds of acute porphyric attacks and subsequent death in heterozygotes without diagnosis and without treatment options [18]. Third, the prognosis of paraplegic patients was abysmal in those times. The normalized SMR after 1965 could be attributed to improved disease recognition, the avoidance of disease triggers, treatment with heme arginate (which was first developed during the 1960s), and family counselling.

Remarkably, the difference in mortality between women and men was not significant. This is remarkable, as women in the general population have lower mortality rates. Women suffered from excess mortality associated with *HMBS* heterozygosity for a longer time period and at younger ages than men. These differences occurred before the widespread use of exogenous sex hormones/oral contraceptives in the early 1960s, which are potential porphyrinogenic triggers [19].

Symptomatic porphyria patients had an SMR of 3.2 (95%CI [2.4–4.0]) relative to age-matched USA 1970 Census death rates [14]. We did not observe excess mortality in the 1970s. The American analysis was restricted to patients hospitalized for acute attacks, whereas we avoided selection based on attacks. The American study emphasizes that patients presenting with symptoms may be at high risk.

However, in a Norwegian cohort study, the risk of premature death was not increased in a cohort with symptomatic patients with any of the acute hepatic porphyrias (i.e., AIP, hereditary coproporphyria, or variegate porphyria) between 1992 and 2017 [20]. Secondly, the results from a Finnish retrospective longitudinal study among female patients living between 1955 and 2015 are consistent with our observations. The study also reported a normalization of life expectancy during the follow-up time compared with the general population. Additionally, they reported a considerable decrease in the incidence of acute attacks starting between 1960 and 1970 [21]. The results of our study, as well as the findings from the Norwegian and Finnish populations, are in contrast with the high mortality observed in historical cohorts [10,11,12,13,14]. Together, these results can be interpreted to underline the significant progress that has been made in porphyria care over the last 40 years.

These findings suggest that the current treatment strategy has reduced excess mortality in specific, susceptible age categories of young women and men. A normalization of mortality ratios is highly relevant for the clinical counselling of asymptomatic HMBS heterozygotes. Asymptomatic individuals can be concerned with questions about life expectancy and what they can expect for their future and that of their family. These individuals can be partly reassured by a normal prognosis following lifestyle regimens. The demonstration of a normal life expectancy can also reduce the impact on life insurance costs and health insurability.

### Strengths and Limitations

The major strength of our observations is that the individuals in this cohort were included without selection based on attacks. With the pedigree SMR method, all observed excess mortality can considered to be related to AIP, as the entire Dutch population was used as a control group. Secondly, the natural history of AIP was studied, with persons living in eras before disease awareness, before diagnostic tests, and without therapeutic options. Thirdly, the pedigrees were identified because the index patient sought medical attention, i.e., all index patients presented with a porphyric attack. However, the overall mortality was equal to that of the general Dutch population. Therefore, one could compare our approach with those of studies based on blood donor cohorts [6], from which we know that the prevalence of the *HMBS* variants is much greater than the occurrence of disease. The increasing use of modern DNA technologies such as exome sequencing increases the necessity of knowing the burden of genetic variants carried by asymptomatic persons

A weakness of our study is that our analyses represent only some of the known pathogenic variants, i.e., 17 out of >400 variants reported in the Human Gene Mutation Database (HGMD). It is likely—but uncertain—that the outcomes of this study could be extrapolated to other *HMBS* gene variants.

Another limitation of our study is that we cannot add information on disease and long-term complication-specific mortality and morbidity (e.g., hypertension, chronic kidney disease, hepatocellular carcinoma). For example, it is known that patients with acute hepatic porphyrias are at increased risk of primary liver cancer (~5% prevalence) compared to the general population (~5 per 100,000 persons) [22], and this is associated with an increased risk of mortality [20]. Because data on causes of mortality could not reliably be collected for most individuals in the dataset, we only analyzed all-cause mortality. Furthermore, the presented results cannot exclude excess mortality in individuals born after 1985, as these subjects have not yet reached the age of late complications. Finally, there were too few patients with recurrent attacks in the dataset to calculate their specific SMR.

In conclusion, overall mortality was not increased in Dutch AIP pedigrees compared to the general population. However, between 1915 and 1964, we observed significant excess mortality in young women and old men. Women had excess mortality at a younger age. During the past half-century, *HMBS* heterozygotes’ life expectancy has normalized.

## Figures and Tables

**Figure 1 life-12-02059-f001:**
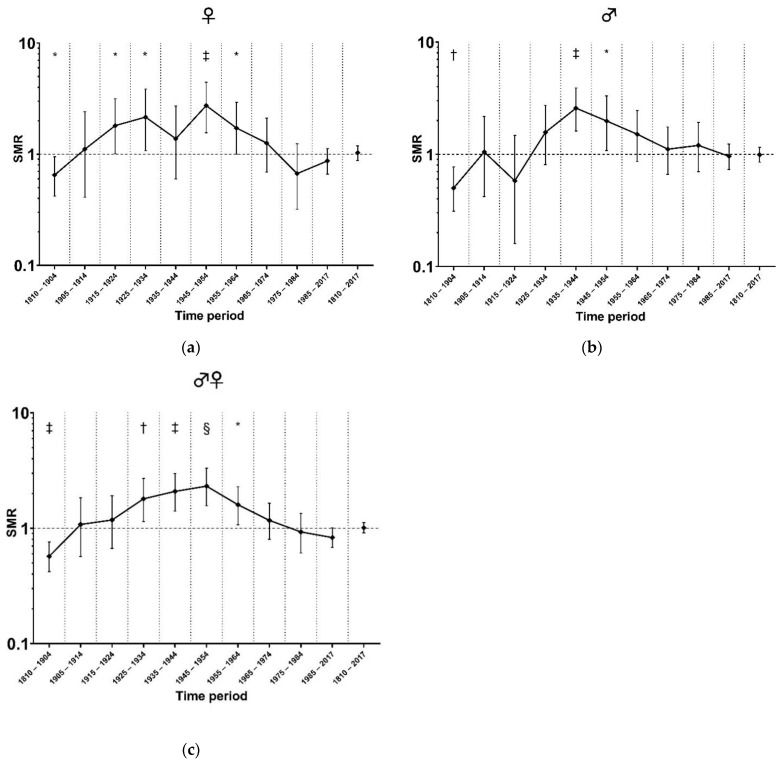
(**a**) SMR per time period (females); (**b**) SMR per time period (males); (**c**) SMR per time period (females and males combined). Abbreviations: AIP, acute intermittent porphyria; SMR, standardized mortality ratio. The *X*-axes depict time periods, while the *Y*-axes are scaled logarithmically. The mortality in Dutch AIP pedigrees in time periods between 1810 and 2017 is expressed as a standardized mortality ratio (SMR) and is depicted as a point estimate for that time interval; 95% confidence intervals are depicted around the point estimates. * *p* < 0.05; † *p* < 0.01; ‡ *p* < 0.001; § *p* < 0.0001.

**Figure 2 life-12-02059-f002:**
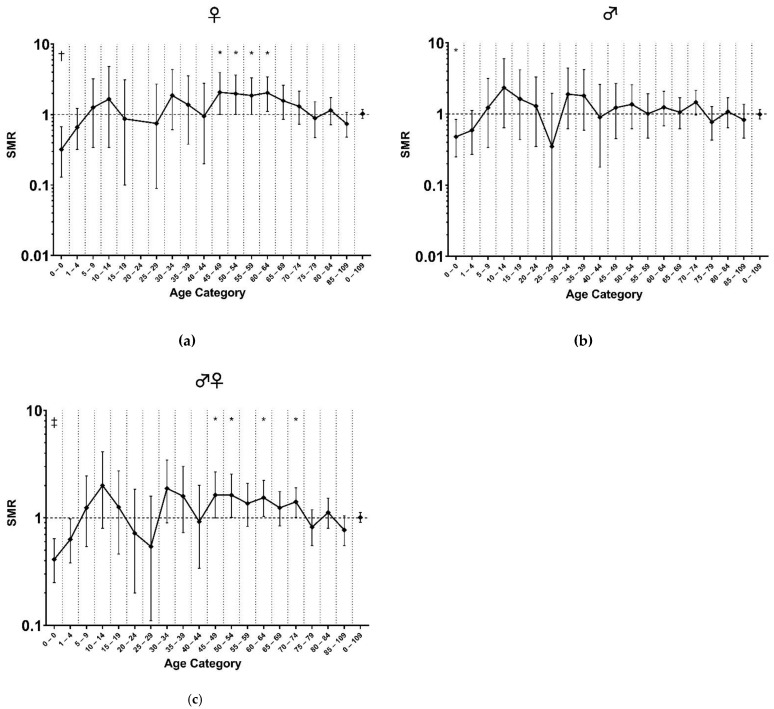
(**a**) SMRs per age category (females); (**b**) SMRs per age category (males); (**c**) SMRs per age category (females and males combined). Abbreviations: SMR, standardized mortality ratio. The X-axes depict different age categories, while the Y-axes are scaled logarithmically. The 95% confidence intervals are depicted around the point estimates. Mortality through the ages is expressed as the standardized mortality ratio (SMR) and is depicted as a point estimate for that age category. * *p* < 0.05; † *p* < 0.01; ‡ *p* < 0.001.

**Table 1 life-12-02059-t001:** Characteristics of the studied cohort of Dutch AIP pedigrees.

	Number (%)
Pedigrees/families included	46
Known *HMBS* variants *	17
Unknown *HMBS* variants	3
Subjects	523
Females	270 (51.6%)
1.0 probability of heterozygosity ^†^	264 (50.5%)
0.5 probability of heterozygosity ^†^	259 (49.5%)
Follow-up time (in PYRS) ^‡^	29,679

Abbreviations: *HMBS*, hydroxymethylbilane synthase; PBGd, porphobilinogen deaminase; PYRS, person-years. * Variants are presented in Table 2. ^†^ The 1.0 probability is based on either the presence of a known pathogenic *HMBS* variant or decreased PBGd enzyme activity; 0.5 probability could be established in first-degree relatives of certain heterozygotes; data collection is described in more detail in the Materials and Methods section. ^‡^ PYRS is the total sum of all years of follow-up of all included persons.

**Table 2 life-12-02059-t002:** All *HMBS* gene variants present in the studied Dutch AIP pedigree cohort.

#	(NM_000190.4)	Number (%)	Number of Individuals	Number of Pedigrees
1	c.346C > T	Missense	125	13
2	c.91G > A	Missense	74	6
3	c.33 + 1G > A	Splicing	68	4
4	c.825 + 1G > A	Splicing	39	1
5	c.181dupG	Frameshift	32	1
6	c.287C > T	Missense	24	2
7	c.500G > A	Missense	23	3
8	c.991G > A	Missense	23	1
9	c.87 + 5G > A	Splicing	21	1
10	c.716A > G	Missense	17	1
11	c.709delG	Frameshift	13	1
12	c.625G > A	Missense	13	1
13	c.219_220delGA	Frameshift	11	1
14	c.499C > T	Missense	11	1
15	c.739T > C	Missense	7	1
16	c.230T > C	Missense	5	1
17	c.275T > C	Missense	5	1
	Unknown ^†^	Missense	16	3

Abbreviations: *HMBS*, hydroxymethylbilane synthase; PBGd, porphobilinogen deaminase. # Previously unpublished variants. ^†^ In these three unrelated pedigrees (up to the year 1810), the *HMBS* gene variant analyses were not performed. The diagnosis of acute intermittent porphyria was based on a decrease in enzyme activity defined as PBGd activity (normal PBGd activity in erythrocytes: >64 pmol/mg protein/h; normal PBGd enzyme activity in lymphocytes: >72 pmol/mg protein/h (since September 2006) or >80 pmol/mg protein/h (before September 2006)).

## Data Availability

Data are available upon request by contacting J.G. Langendonk via stofwisselingsziekten@erasmusmc.nl.

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
