# Peer review of "Mortality in Pedigrees with Acute Intermittent Porphyria"

_life, 2022, doi:10.3390/life12122059_

Round 1
Reviewer 1 Report
The manuscript by Neeleman et al presents an estimation of all-cause mortality rates in families with acute intermittent porphyria (AIP). Since a proportion of patients suffering this condition present high morbidity and mortality this study provides important knowledge and associations with specific time periods and population.
Author’s studies were conducted on patient data from the Porphyria Expert Center (Rotterdam) including individuals with AIP diagnosis. With appropriate exclusions, from an initial population of 1935 individuals from the time period 1810-2017, a total of 523 individuals were selected containing 46 pedigrees, and >26k person-years. Several HBMS variables found in this population are also presented.
Analyzing standardized mortality rates (SMR) in different time periods, authors conclude that peaks values in SMR were observed for women and med in different periods of the last century, although after adjustment not differences in mortality between women and men were observed. Both sexes exhibited significant increases SMR between 1925-1964. Analyses by age group showed a significant higher SMR in women between 45-64 years of age, whereas excess mortality was observed in men 10-74 years old.
The article provides interesting information about the life expectancy of HMBS heterozygous individuals as well as a longitudinal analyses of life-threating effects of living with AIP. An appropriate discussion is presented including limitations, such as the inclusion of only a fraction of known HMBS variants, and the absence of information on diseases or specific morbidity and mortality, these are acknowledged by the authors.
The article is well written and conclusions are supported by results. Quality of figures could be improved, X-axes are hard to read.
Author Response
RESPONSE TO REVIEWER 1
The article is well written and conclusions are supported by results. Quality of figures could be improved, X-axes are hard to read.
Response:
Thank you for the positive review comments. Quality of figures was improved by increasing resolution, increasing font size of the x-axis, enforcing the dotted line (SMR = 1), and adding sex symbols. Also the figures were made more consistent by removing the connecting line between the last data points in all figures.
Reviewer 2 Report
Table 1: states 20 variants
but Table 2: 17 variants plus unknown: this is somewhat confusing.
Figure 1: The year numbers are barely readable: should be enlarged, the dotted line (SMR 1) should be enforced.
Maybe a female gender symbol can be put in Fig 1a, a male gender symbol in 1b and male and female in 1c
Figure 2: same as Figure 1
Patients with porphyria can also die from liver cancer or other mortalities. This has not been mentioned in the discussion.
The reduction in mortality is attributed by the authors to better diagnosis and better counseling. But what about the introduction of heme arginate for the treatment of acute life threatening attacks ?
Author Response
RESPONSE TO REVIEWER 2
Table 1: states 20 variants but Table 2: 17 variants plus unknown: this is somewhat confusing.
Response:
Changes to table 1 and 2 were made to clarify the points made by the reviewer. For table 1, we changed the variants parameter into 2 parameters: known and unknown variants. For table 2 we added information on the unknown variants in the legend.
Figure 1: The year numbers are barely readable: should be enlarged, the dotted line (SMR 1) should be enforced. Maybe a female gender symbol can be put in Fig 1a, a male gender symbol in 1b and male and female in 1c
Response:
Quality of figures was improved by increasing resolution, increasing font size of the x-axis, enforcing the dotted line (SMR = 1), and adding sex symbols. Also the figures were made more consistent by removing the connecting line between the last data points in all figures.
Patients with porphyria can also die from liver cancer or other mortalities. This has not been mentioned in the discussion.
Response:
We have changed the text on page 8, and added a new reference (a recent systematic review on this subject).
The reduction in mortality is attributed by the authors to better diagnosis and better counseling. But what about the introduction of heme arginate for the treatment of acute life threatening attacks ?
Response:
We have moved text from the paragraph on strengths and limitations, to the second paragraph of the discussion and included a discussion on the use of heme arginate.
Reviewer 3 Report
The authors provide a retrospective analysis of AIP morbility in the Netherlands. Although this is a large topic, the authors provide a precise background and criteria for inclusion and exclusion. However, as a long-term historical cause-of-death analysis, it is difficult to accurately record the medical record of every patient with AIP, especially for different families. Moreover, diagnostic testing methods and genetic analysis methods are constantly updated. Although the authors exclude a lot of bias, multiple factors will still interfere with the research. Therefore, I appreciate reading this study, but suggest that the authors try to compress the layout of the article and organize the ideas of the article to make it clearer and more readable.
Author Response
RESPONSE TO REVIEWER 3
Therefore, I appreciate reading this study, but suggest that the authors try to compress the layout of the article and organize the ideas of the article to make it clearer and more readable.
Response:
We have made use of the in-house manuscript editing service of Life in order to make the article clearer and more readable. The layout was improved by improvements made to the figures, and passages were rewritten in the discussion.